# Nondestructive Thickness Measurement of Thermal Barrier Coatings for Turbine Blades by Terahertz Time Domain Spectroscopy

**Longhai Liu [1,2], Haiyuan Yu [3], Chenglong Zheng [1], Dongdong Ye [4,*], Wei He [5], Silei Wang [1], Jining Li [1], Liang Wu [1], Yating Zhang [1], Jianhua Xie [2] and Jianquan Yao [1,6,*]**

1 Key Laboratory of Opto-Electronics Information Technology, Ministry of Education, School of Precision Instruments and Opto-Electronics Engineering, Tianjin University, Tianjin 300072, China
2 Advantest (China) Co., Ltd., Shanghai 201203, China
3 Materials Research Institute, Beijing Beiye Functional Materials Co., Ltd., Beijing 100192, China
4 School of Artificial Intelligence, Anhui Polytechnic University, Wuhu 241000, China
5 Science and Technology on Electromagnetic Scattering Laboratory, Beijing 100854, China
6 Department of Electrical and Electronic Engineering, Southern University of Science and Technology, Shenzhen 518055, China
* Correspondence: jqyao@tju.edu.cn (J.Y.); ddyecust@ahpu.edu.cn (D.Y.)

**Abstract:** Owing to its high penetrability with dielectric materials, terahertz time domain spectroscopy (THz-TDS) is a promising nondestructive measurement technology. The coating thickness deviation and defect of thermal barrier coatings (TBC) will affect its thermal insulation performance and lifetime. In this work, THz-TDS was applied to measure the coating thickness distribution of TBC. The refractive index was obtained by THz-TDS transmission mode. To avoid the normal incidence THz signal loss, the THz signal was reflected from the TBC with a 10° incident angle, which also made the measurement result insensitive to the unevenness and tilt of the TBC sample. In the experiment, the yttria-stabilized zirconia (YSZ) TBC was measured by THz-TDS to estimate the thickness distribution. To validate the thickness measurements, metallography was introduced to correlate the TBC thickness result. The measurement deviation was within 12.1 μm, i.e., 3.45% for the THz-TDS and metallography result. A piece of turbine blade was measured by THz-TDS and a eddy current test. The maximum deviation was 8.48 μm, i.e., 2.36% of these two methods. Unlike the eddy current test, the THz-TDS thickness result was not affected by the cooling holes. The effectiveness of the nondestructive thickness measurement of TBC for turbine blades by THz-TDS was verified.

**Keywords:** terahertz time domain spectroscopy; nondestructive measurement; thermal barrier coatings; turbine blades; yttria-stabilized zirconia

## 1. Introduction

The gas turbine is a high-efficiency thermal energy conversion system widely used in the field of civilian power generation. With the strict restrictions in the emission of environmental pollutants, gas turbines must have the characteristics of high efficiency, high power, and low emissions. The turbine inlet temperature (TIT) of a gas turbine reaches 1500 °C. The structure and materials of the hot-end components become a key factor in supporting the increase in its working temperature [1]. Thermal barrier coating (TBC) technology is currently the only feasible and effective way to increase the gas turbine inlet temperature and energy conversion efficiency of gas turbines and has become one of the key technologies in the manufacture of advanced gas turbines [2]. Typical TBCs consist of three layers, including a superalloy substrate, a bond coat (BC) and a ceramic topcoat (TC). The superalloy substrate is the main frame, and the ceramic topcoat prevents the superalloy substrate from high temperature exposure. Since the conductivity characteristics of the superalloy substrate and TC are different, sometimes the TC will fall from the superalloy

substrate due to thermal stress. To solve this issue, a bond coat is used to bond the substrate and TC layer. Only very few materials can be used as a topcoat of the TBC for combustion chambers and gas turbines. A widely used commercial TBC material is yttria partially stabilized zirconia (PYSZ or YSZ). Only the 8% YSZ can be used for heavy-duty gas turbines, which is also known as 8YSZ [3]. Frequent peak shaving of the grid requires the TBC to possess ultra-high thermal shock characteristics and corrosion resistance. The TBC's ability to resist high temperature depends on its thickness and microstructure [4,5]. Theoretically, TBC temperature resistance performance would be improved when the TBC is thicker. However, a thick TC might fall off the substrate when its interfacial stress also increases accordingly. Thus, the thickness of the TBC must be strictly controlled and precisely measured. In addition, an uneven TBC might fall off after a period of usage. Thickness distribution uniformity is a key factor in TBC lifetime.

In the electromagnetic spectrum, the terahertz (THz) wave frequency range is between 0.1 THz to 10 THz, which lies between the infrared waves and millimeter waves. THz waves own both the characteristics of infrared waves and millimeter waves. Due to the inter-molecular vibrations, biomedical and chemical molecular fingerprint absorption spectrums are in the THz range [6]. THz technology has received increasing attention for biomedical, food and agriculture applications. For example, in biomedical applications, highly sensitive detection of malignant glioma cells, oral cancer cells and lung cancer cells were reported using THz metamaterials and THz time domain spectroscopy (THz-TDS) [7–9]. For food and agriculture application, pesticide detection was enabled by flexible graphene–metamaterial sensors and THZ-TDS [10]. On the other hand, THz waves can penetrate non-conductive materials such as plastic, textile, wood, and paint pigments. THz-TDS can be used to detect objects hidden under these surfaces. For example, a filter capsule was found inside a cigarette [11], inscriptions obscured by time on an early modern lead cross was revealed by THz imaging technology [12]. THz wave technology is thought of as the most promising method to measure non-conductive material thickness. Its usage has been proven in semiconductor chip package and car painting thickness measurements [13,14].

The most-recognized method to measure TBC thickness is via a destructive metallographic microscope, which requires a cross-section to be cut into the sample before measurement [15]. Eddy current testing is a popular nondestructive thickness measurement method. However, it needs to touch the sample and is not convenient for concave shape sample measurement [16]. THz technology was first introduced to measure TBC thickness by Fukuchi et al. [17,18]. A perpendicular incident THz pulse signal was reflected from the top and bottom surface of the topcoat. A beam splitter was used to guarantee perpendicular incidence, and only a one quarter THz pulse was received in the THz detector. Burger et al. reported using the non-perpendicular incidence method to measure TBC thicknesses with reflection THz-TDS [19]. The incident angle was as large as 30°, which is sensitive to the unevenness and tilt of the sample. However, TBC thickness distribution has not yet been reported. In this study, we introduced one method to measure the TBC coating thickness distribution by asynchronous sampling with THz-TDS. The THz measurement head is compact, and the incident angle is as small as 10°, which ensures its robustness for TBC thickness measurement of curved gas turbines samples. The thickness distribution of the TBC sample was obtained, and the thickness of the topcoat of a real turbine blade sample was reported.

## 2. Materials and Methods

A TBC sample (prepared by Beijing Beiye Functional Materials Co., Ltd., Beijing, China) was used as a test sample. The nickel-based superalloy MM247 sample was used as the substrate. A double-layer bond coating was fabricated with commercial NiCoCrAlY powders. The topcoat (TC) layer was fabricated by an atmospheric plasma spraying method using commercial 8YSZ powders.

Asynchronous sampling THz-TDS was introduced to measure the TBC thickness. The block diagram of the THz-TDS system is shown in Figure 1. THz-TDS employs a precise synchronized control unit to control two femtosecond lasers. Laser 1 irradiates the PCA emitter together with electric voltage bias and generates THz pulse, and Laser 2 irradiates the PCA detector to detect THz signal. This technique does not need a mechanical delay stage. By modulating the repeat rate of Laser 1 and Laser 2, a full THz waveform is acquired through interval sampling. To avoid the aberration of the lens, the divergent THz signal is focused on the sample with two pairs of parabolic mirrors instead of a Teflon lens. Transmission mode THz-TDS is used to define the refractive index, while reflection mode THz-TDS is used to measure the TBC sample thickness.

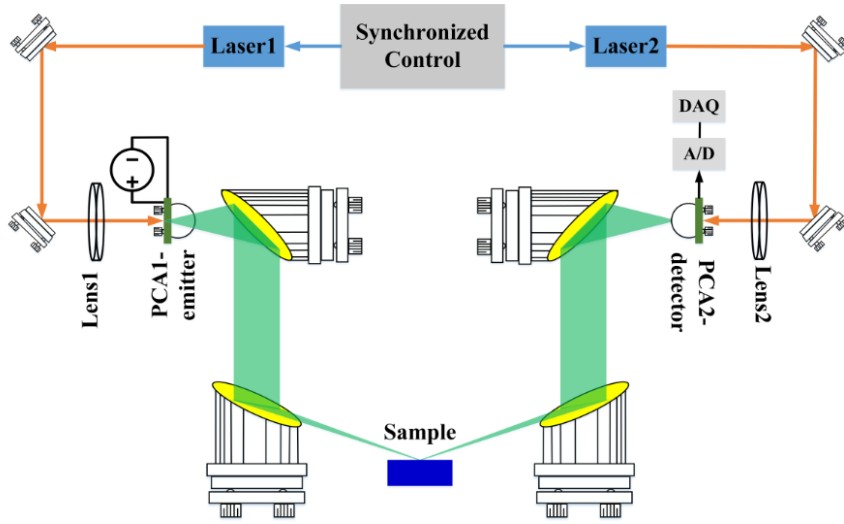

**Figure 1.** The block diagram of THz-TDS system.

In transmission mode, the reference pulse THz signal transmitted through air was detected. Then, after putting the sample in the transmission optics, the detected THz pulse was the sample signal. The refractive index was obtained from the electric field of reference and sample THz signal,

$$\frac{E_S(\omega)}{E_R(\omega)} = A exp(\ i\Phi) = \frac{4n_S(\omega)}{[n_S(\omega)+1]^2} exp\left[-\frac{\alpha_S(\omega)d}{2}\right] \times exp\left\{\frac{i[n_S(\omega)-1]\omega d}{c}\right\} \quad (1)$$

$A$ and $\Phi$ are the magnitude difference and phase shift. $c$ is the light speed, $d$ is preset sample thickness, and $\alpha_S$ is the absorption coefficient. The real part $n_s(\omega)$ and imaginary part $\kappa_s(\omega)$ of the complex refractive index could be expressed as

$$n_S(\omega) = 1 + \frac{\Phi c}{\omega d} \quad (2)$$

$$\kappa_s(\omega) = -\frac{c}{\omega d} \ln\left[\frac{[n_S(\omega)+1]^2}{4n_S(\omega)}A\right] \quad (3)$$

By using a pre-measured thickness 8YSZ topcoat sample, the transmission mode THz-TDS was employed to obtain the refractive index. In the range of 0.2~1.2 THz, the refractive index of 8YSZ was obtained as 5.56.

Reflection mode THz-TDS was introduced to measure the whole TBC sample thickness. When the pulse THz signal was irradiated on the top surface of the TBC, it was partly reflected and partly penetrating the topcoat. The bond coat is the alloy layer, thus the THz wave penetrating the topcoat was totally reflected from the interface between the topcoat and bond coat. When the interface-reflected wave arrived at the topcoat surface, one part of the THz signal transmitted to the outside and the other part was reflected into the topcoat

again to obtain a multiple reflections echo. The reflected THz signal is shown in Figure 2. TBC thickness can be calculated through the time difference between the two reflected THz pulse signals of the top surface.

$$T = \frac{\Delta t \, c \, cos\theta}{2n} \tag{4}$$

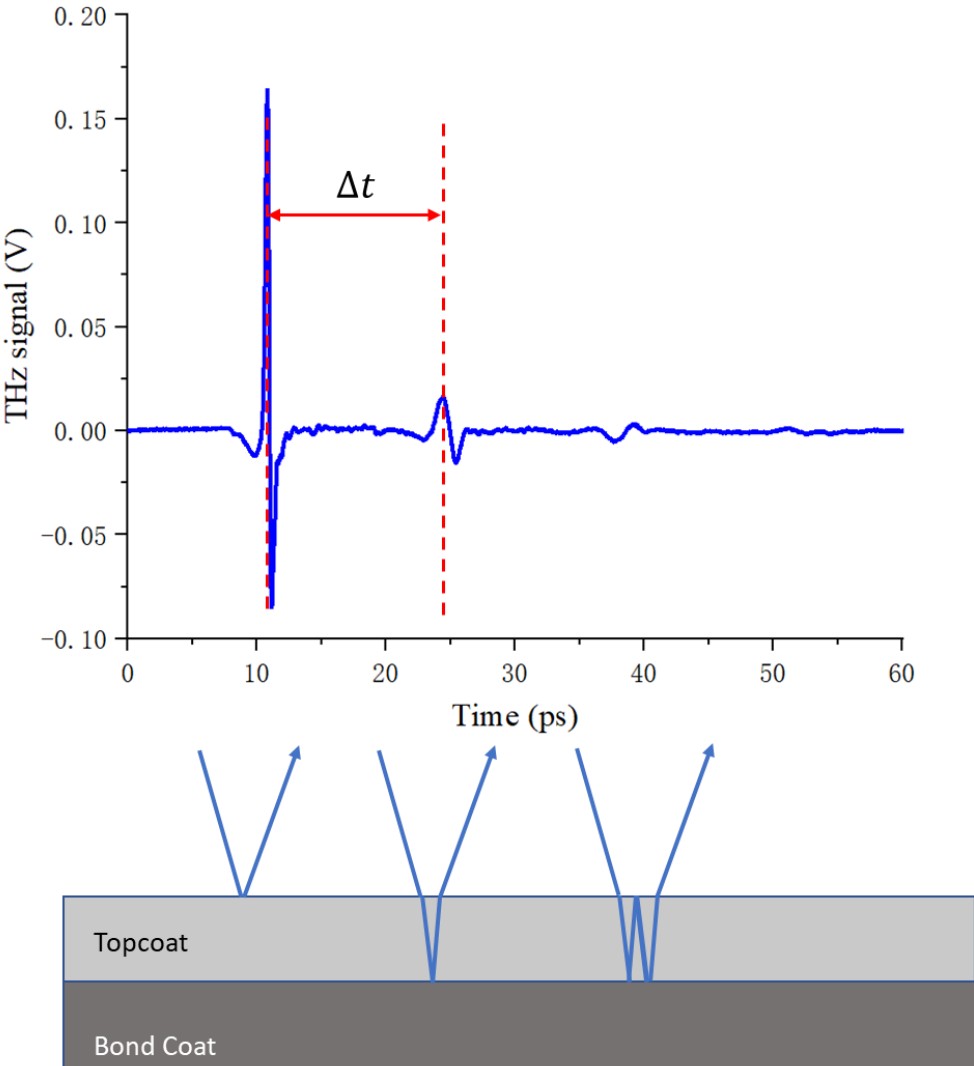

**Figure 2.** THz signal reflected from the topcoat surface and the interface of topcoat and bond coat.

While $\Delta t$ is the time difference between the two reflected pulse, $\theta$ is incident angle of THz signal, and $n$ is the refractive index of the topcoat materials. To evaluate the effect of the incident angle and calculated thickness, it could be expressed as

$$T' = -\frac{\Delta t \, c \, sin\theta'}{2n} \tag{5}$$

It is obvious that the smaller incident angle corresponds to the more accurate measurement, thus small incident angle reflection THz optics was used in the reflection THz-TDS system.

## 3. Results

A commercial THz-TDS (TAS7500TS, Advantest Corporation, Tokyo, Japan) was used to measure the TBC thickness in the experiment. The frequency range was 0.1~4 THz, and the time domain period was 131 ps. The macroscopic photo of the real measurement of

the THz optics is shown in Figure 3. The THz signal was reflected from the TBC topcoat with an incident angle of 10°. The 2D scanning stage was synchronized with the THz signal to obtain the whole TBC thickness distribution. The maximum 2D scanning range was 25.6 mm × 25.6 mm, and the scanning resolution was set as 0.8 mm. THz waveforms were averaged 1024 times to reduce the sampling noise. In addition, to eliminate water vapor absorption, dry compressed air was used to purge the whole THz optics during the measurements. By utilizing the above method, the whole TBC topcoat thickness distribution was obtained, and is shown in Figure 3 in the 2D and 3D illustrations. The average thickness was around 379.97 µm, and the thickness deviation was 9.30 µm. Results showed that the TBC sample thickness had a tendency to be concave in the middle.

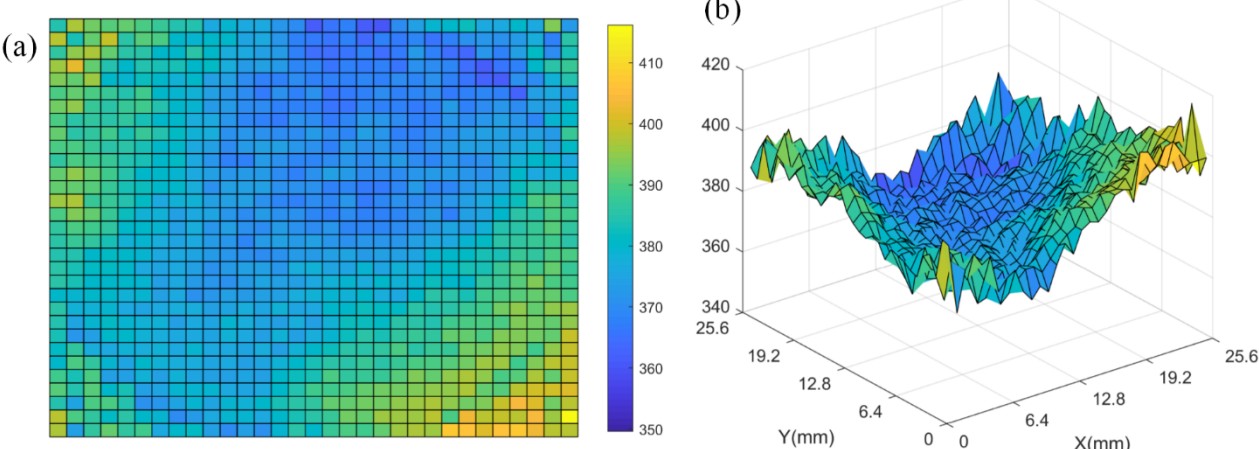

**Figure 3.** TBC thickness distribution measured by THz-TDS: (**a**) 2D thickness distribution; and (**b**) 3D thickness distribution.

To correlate the THz measured thickness, the TBC sample was cut as a cross-section and measured by a metallographic microscope. The middle, thinner positions were marked with a black line at the TBC back side to ensure that the same positions were measured by the THz signal and the metallographic microscope. One of the metallographic microscope images is shown in Figure 4.

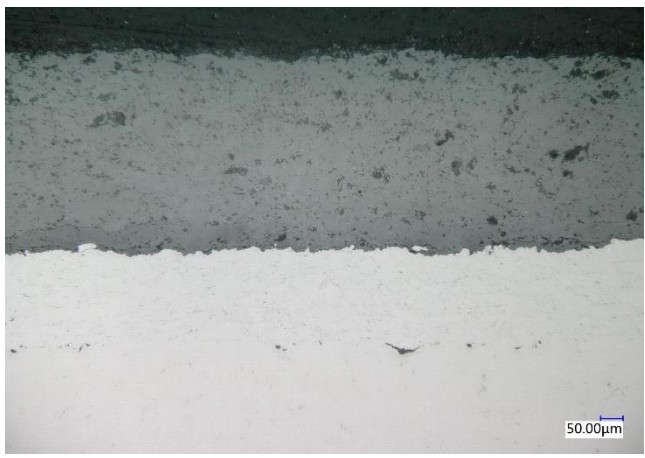

**Figure 4.** Metallographic microscope image of cut TBC sample.

The metallographic microscope images were analyzed by its imaging software. Thicknesses were determined by visually averaging the distance between the air–topcoat and topcoat–bond coating interfaces. Since the THz spot size was around 300 µm, three points with intervals of 100 µm were averaged as the final thickness result. The correlation results

are presented in Table 1. Due to the unevenness of the TBC surface, human operation errors occurred during the metallographic microscope thickness measurement. The maximum thickness deviation was 12.1 μm, i.e., a 3.45% deviation for the experimental THz-TDS measurement data compared to the metallography reference measurement data. The THz-TDS measurement result was validated.

**Table 1.** The correlation of thickness measurement result by THz-TDS and metallographic microscope.

| Measured Position | Thickness by THz-TDS (d1)/μm | Thickness by Metallographic Microscope (d2)/μm | Deviation (d1–d2)/μm | Deviation (d1–d2)/d2 |
|---|---|---|---|---|
| Position 1 | 378.8 | 386 | −7.2 | −1.86% |
| Position 2 | 372.4 | 370 | 2.4 | 0.65% |
| Position 3 | 362.1 | 350 | 12.1 | 3.45% |
| Position 4 | 352.6 | 350 | 2.6 | 0.74% |
| Position 5 | 346.6 | 340 | 6.6 | 1.94% |

To extend the engineering application, a small piece of turbine blade was measured by the reflection THz-TDS optics. Since the piece of turbine blade had a curved shape, it was difficult to scan the surface with the above 2D scanning stage. As indicated in Figure 5, the measurement locations were marked on the side section surface, while the center of the topcoat surface was measured by THz-TDS. Nine positions of the concave surface were measured. Since this piece of turbine blade was as large as 120 mm, it was difficult to prepare the metallographic sample. Thus, the eddy current test (MiniTest 600, ElektroPhysik, Köln, Germany) was introduced to measure the curved topcoat thickness. Eddy current testing measured the change of the induced eddy current in the inspected sample to non-destructively evaluate the thickness of the turbine blade sample. In position 6, the cooling holes affected the distribution of electromagnetic induction, which then affected the eddy current measurement accuracy. Thus, the eddy current could not precisely measure the topcoat thickness. The comparison of THz-TDS and eddy current test result are shown in Table 2. Except for position 6, good agreement between the two results were indicated. The maximum deviation was 8.48 μm, i.e., a 2.36% deviation between the two results. To measure the whole curve turbine blades thickness distribution, collaboration with a 6-axis robot system is needed to ensure that the THz pulse signal is perfectly incident on the turbine blade's surface.

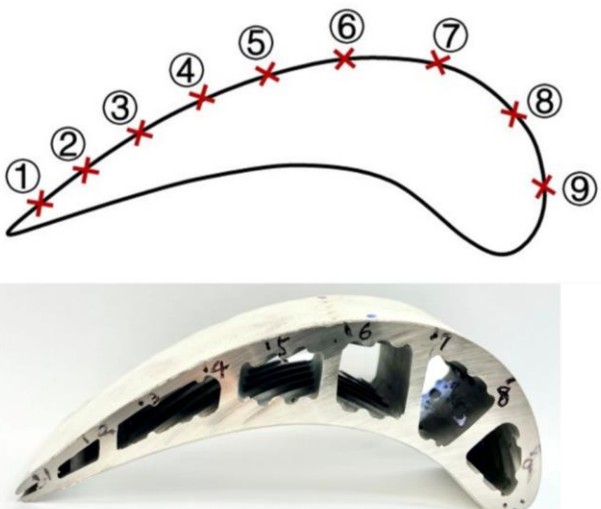

**Figure 5.** Schematic drawing and macroscopic photo of a piece of turbine blade.

**Table 2.** The correlation of thickness measurement result of turbine blade by THz-TDS and the eddy current test.

| Measured Position | Thickness by THz-TDS (d1)/μm | Thickness by Eddy Current Test (d2)/μm | Deviation (d1–d2)/μm | Deviation (d1–d2)/d2 |
|---|---|---|---|---|
| Position 1 | 361.55 | 366 | 4.45 | 1.22% |
| Position 2 | 374.51 | 370 | −4.51 | −1.22% |
| Position 3 | 375.47 | 377 | 1.53 | 0.41% |
| Position 4 | 374.62 | 380 | 5.38 | 1.42% |
| Position 5 | 333.83 | 330 | −3.83 | −1.16% |
| Position 6 | 375.74 | 340 | −35.74 | — |
| Position 7 | 430.66 | 430 | −0.66 | −0.15% |
| Position 8 | 352.47 | 350 | −2.47 | −0.71% |
| Position 9 | 351.52 | 360 | 8.48 | 2.36% |

## 4. Conclusions

In this paper, we presented a nondestructive measurement method for thermal barrier coatings for turbine blades using reflection THz-TDS. The pulse THz signal was reflected from the surface of the topcoat and the interface of the topcoat and bond coat of the TBC. By calculating the time difference, the TBC topcoat thickness was obtained. A 2D scanning stage was used with the TBC sample. The TBC topcoat thickness distribution was remapped. In the experiment, the TBC sample thickness was measured by THz-TDS and a metallographic microscope. The maximum thickness deviation between the two measurements was 12.1 μm, i.e., 3.45%. A piece of turbine blade thickness was measured by THz-TDS and a eddy current test. Except for the cooling holes eddy current test error, the maximum thickness deviation was 8.48 μm, i.e., a 2.36% deviation between the two methods. In further work, in order to obtain the whole thickness distribution of the curved turbine blade, a 6-axis robot system will be introduced to ensure that the THz signal is perfectly incident on the turbine blade sample with the target incident angle.

**Author Contributions:** Conceptualization, L.L. and H.Y.; methodology, L.L.; software, C.Z.; validation, L.L. and H.Y.; formal analysis, C.Z.; investigation, S.W.; resources, D.Y.; data curation, J.L.; writing—original draft preparation, L.L.; writing—review and editing, W.H. and D.Y.; visualization, Y.Z.; supervision, L.W.; project administration, J.X.; funding acquisition, J.Y. and D.Y. All authors have read and agreed to the published version of the manuscript.

**Funding:** This research was funded by the National Key Research and Development Program of China (2017YFA0700202), and National Natural Science Foundation of China (61735010, 52205547), Basic Research Program of Shenzhen (JCYJ20170412154447469), Key Research and Development Projects in Anhui Province (2022a05020004).

**Institutional Review Board Statement:** Not applicable.

**Informed Consent Statement:** Not applicable.

**Data Availability Statement:** Not applicable.

**Conflicts of Interest:** The authors declare no conflict of interest.

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
