# Peer review of "Nondestructive Thickness Measurement of Thermal Barrier Coatings for Turbine Blades by Terahertz Time Domain Spectroscopy"

_photonics, doi:10.3390/photonics10020105_

Round 1

Reviewer 1 Report

This paper is high quality, some minor comments:

(1)  For actual turbine blade detections, how to ensure the THz waves with 10° incident angle?

(2)  What is the accuracy of the eddy current system? Could you provide the product type?

(3)  For Fig.1, please give the amplitude of the THz signal.

(4)  In Table 2, the reasons for the large difference between results of THz and eddy current in Position 6 should be given in detail.

(5)  Please use blank space in the following cases: 0.8 mm, 25.6 mm etc.

Reviewer 2 Report

The authors used THz-TDS to perform the nondestructive thickness measurement of TBC for turbine blades. Some issues should be improved before published.

1 In Line 80, ‘it’ should be ‘It’. In Line 123, ‘the’ should be ‘The’.

2. In Line 93, the last sentence is incomplete.

3. In Line 123, the authors claimed that the refractive index of 8YSZ is 5.56. It is better to give the plots of the refractive index measured results, and the frequency range should be given.

4.In Line 167, the sentence ‘The middle thinner positions were marked with redline at the TBC back side…’. It can not be reflected in Figure 4.

5 In table 1, the difference between the deviations is large, such as 0.65% and 3.45%, and the results need to be discussed in detail.

Author Response

the reference is also updated.
